# Tuberculosis as a Risk Factor for 1918 Influenza Pandemic Outcomes

**DOI:** 10.3390/tropicalmed4020074

**Published:** 2019-04-29

**Authors:** Svenn-Erik Mamelund, Jessica Dimka

**Affiliations:** 1Work Research Institute, OsloMet—Oslo Metropolitan University, PO. Box 4, St. Olavs Plass, 0130 Oslo, Norway; 2Department of Anthropology, University of Pittsburgh, Pittsburgh, PA 15260, USA; jld199@pitt.edu

**Keywords:** 1918 pandemic, Spanish flu, tuberculosis and influenza interactions, morbidity, case fatality, case-control studies, historical epidemiology

## Abstract

Tuberculosis (TB) mortality declined after the 1918 pandemic, suggesting that influenza killed those who would have died from TB. Few studies have analyzed TB as a direct risk factor for 1918 influenza morbidity and mortality by age and sex. We study the impacts of TB on influenza-like illness (% of population sick) and case fatality (% of cases dying) by age and sex through case-control comparisons of patients (*N* = 201) and employees (*N* = 97) from two Norwegian sanatoriums. Female patients, patients at Landeskogen sanatorium, and patients aged 10–39 years had significantly lower morbidity than the controls. None of the 62 sick employees died, while 15 of 84 sick patients did. The case-control difference in case fatality by sex was only significant for females at Lyster sanatorium and females at both sanatoriums combined. Non-significant case-control differences in case fatality for males were likely due to small samples. Patients 20–29 years for both sexes combined at Lyster sanatorium and at both sanatoriums combined, as well as females 20–29 years for both sanatoriums combined, had significantly higher case fatality. We conclude that TB was associated with higher case fatality, but morbidity was lower for patients than for employees. The results add to the study of interactions between bacterial and viral diseases and are relevant in preparing for pandemics in TB endemic areas.

## 1. Introduction

Killing an estimated 50–100 million people worldwide, the 1918 pandemic has long been recognized as a syndemic, in which synergistic conditions, such as co-infection with bacterial pathogens, exacerbate outcomes, within the context of social inequalities and chronic stresses [1,2,3,4]. Even today, the WHO International Classification of Diseases groups pneumonia and influenza (PI) into the same category [5], while much research has focused on the links between influenza and pulmonary *Mycobacterium tuberculosis* (TB). A chronic, slow bacterial disease, TB can damage lung tissue, creating additional surfaces for other respiratory pathogens to attach, and impair recovery and the immune system [6,7,8,9,10]. Further, mouse models and other studies show that both diseases can impair immune system responses, promoting subsequent infection, enhanced mycobacterial growth, or the emergence of active TB in individuals with latent infections [11,12,13]. Reports of the 2009 H1N1 pandemic from South Africa show that TB was associated with 10% of deaths despite an overall prevalence of only 1%, indicating that TB is a risk factor for influenza mortality [14]. Other studies have produced mixed results, however; for example, de Paus et al. [15] did not find a correlation between the presence of influenza A antibodies and the development of TB, although antibody titers were enhanced, suggesting recent infection.

Tuberculosis had begun to steadily decline by 1800, albeit with peaks during the 1889 and 1918 influenza pandemics, due to factors such as improvement in living conditions and diet, the reduction of effective contacts with infectious cases, and natural selection [10,16,17,18]. The increase in TB mortality associated with World War I and the 1918 pandemic reflects a temporary reversal in the trend, likely due to conditions including poor ventilation, overcrowding, a shortage of medical care, and increased rates of conversion from latent to active infection [16].

Despite the downward trend, TB was still a widespread condition in the early 20th century (e.g., [8]). Even with lower rates today, an estimated 10 million people became ill with TB in 2017, with rates particularly high in developing countries [19]. Influenza pandemics occur 3–4 times each century. After 1918, there were pandemics in 1957, 1968 and 2009. Most experts predict another influenza pandemic is not a matter of if, but when [20,21], so research on historical experiences, including the 1918 pandemic, will provide vital insights to contemporary public health concerns. In particular, the demonstration of increased risk of case fatality (% of cases dying) among TB patients would inform surveillance, vaccination, and treatment policies and practices. However, much work on the role of TB during the 1918 pandemic has focused on subsequent outcomes such as life expectancy, while TB as a risk factor needs further study.

Observed differences in post-pandemic demographic trends prompted Noymer and Garenne [6] to rediscover a connection between the 1918 pandemic and subsequent life expectancy measures among TB patients, similar to earlier observations by Abbott [22] (cited in [23]). The selective mortality hypothesis argues that the pandemic played a role in the early 20th century decline in TB by killing individuals who would otherwise have died of TB at a much later date. This process resulted in longer life expectancies for affected populations. In the United States, the female advantage in life expectancy relative to males, among whom TB was more prevalent, was reduced even into the 1930s. Noymer [24] later proposed two potential mechanisms for this selection. Active selection would indicate physiological vulnerability to death from influenza, or the 1918 strain in particular, as a direct consequence of infection with TB. On the other hand, a weaker, passive selection would be possible with high rates of flu deaths among age groups with TB, even if there was not a higher risk of mortality associated with the disease.

Subsequent research has included retests or theoretical consideration of this hypothesis with different geographic regions and data sources. Many of these studies have explored changes or correlations in mortality and/or life expectancy rates during or after the pandemic, with the assumption that these patterns reflect higher deaths among TB sufferers due to either active or passive mechanisms [8,23,25,26,27]. Yet, Tripp and Sawchuk [17] disagreed that the pandemic played a singular or key role in the decline in tuberculosis. As noted above, the downward trend was already established, and a “harvesting” effect of complications due to underlying health conditions implies not only a short-term impact but a return to background levels rather than a persistent decline.

Although contemporary scholars noted at the time of the 1918 pandemic that mortality was linked to conditions including secondary pneumonia and TB (e.g., [22,28,29]), fewer studies have explored whether individuals with TB did, in fact, suffer higher rates of morbidity and mortality compared to control groups. A review of the contemporary data and literature from several TB sanatoriums (patients vs. employees) and non-institutionalized populations (patients and healthy controls living in same households) in the United States did not document systematic differences in morbidity; statistical tests were not employed to check whether the mostly small differences were significant, and sometimes morbidity was higher for TB cases and sometimes higher for healthy controls [30]. However, in a re-analysis of individual-level historical household data from the US, Noymer and Garenne [6] found that when controlling for household size, TB-infected persons were 2.2 times more likely to contract influenza *outside the household* than non-TB persons living in the same household in 1918. Oei and Nishiura [31] analyzed a Swiss TB sanatorium with 102 TB patients and 33 employees and found that 62.7% of the patients and 72.7% of the employees suffered from influenza-like illness (ILI). In this institutional setting, morbidity was thus lower for the patients than for the employees. The authors found a marginal association (Fisher’s exact, *P* = 0.09) between TB and influenza case fatality risk when comparing patients and employees, as 7 of 64 TB patients with influenza died (10.94% case fatality) while none of the 24 controls with influenza did so (0% case fatality). Additionally, young adults were known to be at highest risk of having latent TB in 1918, which may be among the reasons why positive excess influenza mortality mainly occurred among those aged 20–40 years [31]. Studies from more recent seasonal influenza epidemics reinforce these findings, including analyses of hospitalized patients in South Africa from June 2010 to December 2011. However, this study only found an association between co-infection and increased risk of death in patients who had influenza symptoms ≥ 7 days [32]. A study of the differential impact of the 2009 pandemic also showed that TB was a risk factor for developing a severe disease [33]. 

In this paper, we add to the literature in two ways, as few studies have had data on ILI cases in addition to ILI deaths by age and sex. First, we compare not only case fatality (% of cases dying) but also morbidity among patients and staff at two TB sanatoriums in Norway. Comparing the differences in case fatality rather than mortality (deaths/population) enables us to control for the observed differential risk of developing an ILI given exposure. We thus contribute to the understanding of the circumstances leading to worse outcomes for TB patients. Second, we add to the literature by being the first to study smaller subsamples and thus the role of age and sex in the association between tuberculosis and 1918 influenza outcomes. These analyses are made possible by a sample size that is 2–3 times larger than prior studies analyzing data from TB sanatoriums, such as Oei and Nishiura [31], who had data for only one sanatorium with 102 TB patients and 33 employees. In our study, we have data for two sanatoriums and report results by age and sex for 201 TB patients and 97 employees.

Using a framework to analyze the risk factors leading to unequal pandemic outcomes developed by Crouse and Supriya [34], we assume that (A) the TB patients and the healthy controls had similar risks of exposure to the virus after it passed the doorstep of the institutions. This would be due to no differences in (1) crowding and assortative mixing; (2) access to water and handwashing behavior; and (3) ability to engage in social distancing behavior. However, as the TB patients may have had impaired immune systems [6,7,8,9,10,11,12,13], we assume that (B) TB patients were more susceptible to developing influenza once exposed and more likely to experience complications like pneumonia. We also assume that (C) the quality of basic health care (water, food, bed, and emotional support) offered to and received by the TB patients and employees were the same. As there were no effective antivirals, vaccines, or antibiotics in 1918, medical treatment options were equally ineffective for patients and employees. Therefore, our expectations were that there was either no case-control difference in morbidity or higher morbidity for the TB patients, and that case fatality was higher for the TB patients compared to the employee controls. Our main finding is that TB was associated with a higher risk of death from influenza, but that in these institutional settings, morbidity was lower for TB patients than for employees.

## 2. Materials and Methods 

### 2.1. Sample Details

We use published aggregate-level data on cases of influenza-like illness (ILI) and ILI deaths by age and sex from patients and employees at Lyster and Landeskogen TB sanatoriums in Norway in the fall of 1918. These data are from a report by Olav Hanssen who also presented information on the 1918 pandemic for the city of Bergen and other areas of southwestern Norway using data from a population-representative survey, firms, schools, boats, lighthouses, and mental asylums [29]. The annual reports from the sanatoriums did not normally provide such detailed morbidity and mortality data for the patients, and data on morbidity and mortality were never given for the employees in these reports. The data used here were produced by the chief medical doctors at the respective sanatoriums as part of a direct request by Hanssen. The sanatoriums analyzed in this paper are fairly representative of such institutions in Norway in the early 20th century. For example, the number of beds ranged from fewer than 10 to over 200. In comparison, Lyster and Landeskogen, which were 2 of 6 state sanatoriums, both had a capacity of around 120 beds in 1918. While some sanatoriums were only open to children, or to the rich at privately-owned sanatoriums, Lyster and Landeskogen were public and open to patients of all age groups and socioeconomic subsets of society [35]. We do not have information on the illness onset, illness severity, or the durations of patient stays at the sanatoriums. However, it is reasonable to believe that those institutionalized were sicker than the non-institutionalized suffering from tuberculosis. 

Lyster sanatorium, located in the county of Sogn and Fjordane on the west coast, had 96 beds when it opened in 1902, but the capacity increased to 120 beds in 1924. During the pandemic in 1918, there were 128 patients (69 males, 59 females) and 62 employees (17 males, 45 females) at Lyster. In other words, there were 2.1 patients per employee, and the hospital had slightly more patients than they had beds [29]. Despite being over capacity, the building was huge at 5000 m^2^ and gave 26 m^2^ of floor space per person. The hospital was located on the mountainside, 500 meters above sea level, above Lyster. From the steamship-quay in Lyster, there was an electric cableway up to the sanatorium. A seven-kilometer road with thirteen hairpin turns also led up to the hospital. Due to its location and the infection risks, the institution built up a self-sufficient community with electricity from a private water turbine, residential houses for the doctors and nurses, two family houses for stokers and gardeners, a chapel with a morgue, post office, laundry, icehouse, stables, and a pig farm [35].

Landeskogen sanatorium was located in Grendi on the east side of the Bygland Fjord in the county of East-Agder, a rather remote area of southern Norway. When the hospital opened in 1916 it had 120 beds, but in 1926, the capacity was 134 beds. In 1918, there were 73 patients (39 males, 34 females) and 39 employees (9 males, 30 females), that is 1.9 patients per employee. Only 60.8% of the 1916 bed capacity at Landeskogen was used during the pandemic [29].

The data are available in seven 10-year age groups (10–19,…,70–79) by sex for both patients and employees at each sanatorium. There were 201 patients (84 ILI cases, 15 deaths) and 101 employees (66 ILI cases, 0 deaths). As none of the patients at the two sanatoriums were older than 59 years, 2 male employees aged 60–69 years, and 1 male and 1 female employee aged 70–79 years, all at Lyster, were removed from all analyses to make the cases as equal to the controls as possible. Out of these four employees, only the woman aged 70–79 had an ILI case. Additionally, 3 doubtful cases of ILI among employees living outside Landeskogen sanatorium (2 females aged 20–29 years and 1 female aged 30–39) were removed from case analyses but were retained in the population at risk. The employee sample used in analyses was thus 97 individuals (62 ILI cases). In order to analyze the data, we transformed the aggregate-level data into individual-level data for the cases and controls aged 10–59 years with a set of dummy variables with information on ILI cases, ILI deaths, TB-status, age, sex, and sanatorium. 

### 2.2. Analyses

The analysis proceeds in two steps. First, descriptive statistics are presented comparing the distribution of age, sex, and sanatorium residence, and the distribution of ILI cases (% of pop) and ILI deaths (% of cases) by TB status. Second, the importance of TB status for individual variation in ILI is examined. Since the outcome variable is binary, we use logistic regression models to perform the analyses, with a statistical significance level of 5 percent. To uncover any spurious relationship, and to investigate whether other independent variables can explain the relationship between morbidity and TB, we gradually include control variables in three steps. The first model examines the bivariate relationship between ILI and TB status. To check whether differences in TB status can be explained by sociodemographic variables, we successively add controls in the second to the fourth model for age, sex, and sanatorium. In addition, we perform separate ILI models for each sex, age, and sanatorium group, where the other available covariates are also included. In other words, in these models, (1) each sex interacts with TB status, age, and sanatorium; (2) each age group interacts with TB status, sex, and sanatorium, and (3) each sanatorium interacts with TB status, age, and sex.

Since no deaths occurred among the healthy employee controls, we could not perform similar multivariate regression models for case fatality (% of cases dying). Further, the sample size is also dramatically reduced when the risk group is the ILI cases and not the sample population. Therefore, to analyze the role of TB in the conditional risk of PI death among the ILI cases, we tabulated 2 × 2 cross tables. We also did this for various trivariate interactions of age, sex, and sanatorium. As mentioned above, by considering case fatality rather than mortality (deaths/population), we control for the observed differential risk of developing an ILI given exposure. Our mortality analyses should be interpreted with caution due to the small sample size. However, to take statistical uncertainty into account, we test the associations using a 2-sided Chi-square test or a Fisher’s exact test (as in [31], which had an even smaller sample size). The 2-sided Fisher’s exact test is designed to test for statistically significant differences in extremely small samples. We use this test when one or several of the cells in the 2 x 2 cross tables has an expected value less than 5. In order to be fully transparent with our results, we present each cell even if they contain the value 0 (no ILI cases or deaths) or a “-”, meaning that there were no TB patients or employees in an age, sex, or sanatorium category. 

## 3. Results

### 3.1. Descriptive Findings

Table 1 shows the distribution of age, sex, and sanatorium by case-control status for the total sample. On average, TB patients in the two sanatoriums are 5 years younger than the healthy employee controls. The age difference between patients and employees is 7 years at Lyster (24.06 years vs. 31.03 years) and 2 years at Landeskogen (27.60 years vs. 29.36 year). While more TB patients are male (53.7%) than female (46.3%), there is a strong sex bias among the employees where more than 3 out of 4 are females, most likely because most of the staff would be nurses, a gendered occupation particularly at the time. These sex differences are consistent within each of the two institutions (results not shown). Finally, approximately 60% of patients and controls are from Lyster sanatorium.

Table 2 shows that morbidity in the total sample is higher for the employees than for the TB patients, and that this pattern holds for those aged 10–39 years, for females and at both Lyster and Landeskogen. For males and for 40+ age groups, however, morbidity is higher for TB patients than for employees. Table 2 also shows that there are no deaths among the employees, and that all deaths among the TB patients, except for one death in the age group 50–59, occur in ages 10–29 years. Of the 15 deaths, 13 are from Lyster. Case fatality is 18% overall, with relatively small sub-group differences.

### 3.2. First Cases and Overall Morbidity by Sanatorium

As in most countries, the 1918–1919 pandemic also came in three waves in Norway. The first occurred in June–August, the second in October–December, and the third started around the turn of the year and ended in the late winter and spring of 1919 [36]. Although the first influenza wave was widespread in the village of Lyster during the summer of 1918, only a farm boy came down with influenza at Lyster sanatorium; the patients and other employees escaped as the farm boy was isolated. However, in the beginning of October, during the second wave in Norway, seven new TB patients, who arrived in Lyster by steamboat, brought the disease up to the institution. Subsequently, more than half of the patients (53.1%) and employees (55.2%) became ill. With the exception of two cases of ILI at the end of November 1918, the main outbreak at the institution was over by the end of October. The fall wave came to the non-institutionalized population down in the village of Lyster one month after it started at the sanatorium, at the beginning of November [29]. 

At Landeskogen Sanatorium, the first outbreak also started during the second national wave during the fall of 1918; the first cases were reported on December 2 when 2 of the employees fell ill [29]. These employees were probably infected due to contact with other members of the non-institutionalized population in Bygland, where the first outbreak of the pandemic started on November 18 [36]. After 8–10 days, 77% of the employees at the sanatorium had ILI. The sick employees were isolated, but after 14 days, the patients also started to become sick. Male patients were sick one month before the first female patients were [29]. The influenza morbidity rate for TB patients was 22%.

### 3.3. Multivariate Analysis of Morbidity

Table 3 presents univariate (model 1) and multivariate models (models 2–4) testing the associations between ILI, TB status, and the other covariates. Model 1 shows that morbidity is significantly lower among the TB patients. Controlling for age in model 2 substantially reduces the odds ratio for TB patients (from 0.41 to 0.25) because patients are younger and so more represented in age groups with higher morbidity. Subsequent controls for sex and sanatorium in models 3 and 4 marginally reduce the odds ratios for TB patients, and in model 4, with all other factors the same, the morbidity is 88% lower than the employee rate. The significantly lower morbidity of the TB patients in model 1 was not “explained” away by gradually including more and more covariates. In other words, the association between TB and morbidity is not spurious. Finally, the included measures of model fit mainly improve going from model 1 to models 2, 3 and 4. Model 4 gives the best model fit with the lowest values on AIC, SC and −2 Log L.

In model 4, we also see that 20–29-year-olds have significantly higher morbidity and that 30–39 and 50–59-year-olds have lower morbidity, and that patients at Lyster have significantly higher morbidity. 

We have also performed separate analyses of ILI where sex, age, and sanatorium interact with TB status (not shown). First, separate models for sex show that only female TB patients have significantly lower morbidity than the female employee controls, when controlling for age and sanatorium (OR 0.13, 95% CI 0.06–0.29). One reason for the lack of significance for male TB status is that there are few male employees. Second, the lower ILI rates for TB patients only remain for ages 10-39 years, when controlling for sex and sanatorium (10–19 years: OR 0.20, 95% 0.05–0.89; 20–29 years: OR 0.20, 95% CI 0.08–0.50; 30–39 years: OR 0.15, 95% CI 0.03–0.71). The validity of models for age groups older than 40 years is questionable due to the few cases. Third, separate models for sanatorium show that Landeskogen has significantly lower morbidity for TB patients, when controlling for age and sex (OR 0.03, 95% CI 0.01–0.14).

### 3.4. Case Fatality by Sex and Sanatorium

Here, we present results for case fatality by sex and sanatorium. The associations are tested using a 2-sided Chi-square test or Fisher’s exact (as in [31]). As discussed above, the 2-sided Fisher’s exact test is used when one or several of the cells in the 2 × 2 cross tables has an expected value less than 5. There are no deaths among the employees. Among patients, there are 6 male and 7 female deaths at Lyster. All of these deaths were attributed to pneumonia and occurred among patients with advanced TB. The case fatality is 16.2% among the male patients and 22.6% among the female patients. Only female TB patients have a significantly higher case fatality than the female controls (*P* = 0.012, see Table 4a). However, the lack of a significant finding for males is probably due to few male employee cases. Two male patients died at Landeskogen (15.4% case fatality, ns. difference vs. controls), but no deaths among female patients or employees were reported (Table 4b). 

When considering both sanatoriums combined, the case fatality of female TB patients (20.6%) is significantly higher than for the female employee controls (*P* = 0.001). The case fatality of male TB patients at the two sanatoriums combined (16%) is not significantly different from that of the healthy controls (Table 4c); again, this is probably due to few male employees and few male employee ILI cases. 

### 3.5. Case Fatality by Age, Sex, and Sanatorium

In Table 5a–c, we present analyses on case fatality by age and sex within Lyster and Landeskogen TB sanatoriums and for the two sanatoriums combined, using the same Chi-square or Fisher’s exact strategy described above. With the exception of 1 male death of a patient aged 50–59 at Lyster sanatorium, all of the deaths are among those aged 10–19 and 20–29 years. Due to few cases and deaths, TB has a marginally significant impact on case fatality for the 20–29-year age-group for both sexes combined at Lyster sanatorium (*P* = 0.095 for a case fatality of 23.8%) and at the two sanatoriums combined (*P* = 0.003 for a case fatality of 19.6%). We also find a significant association between TB and influenza for females aged 20–29 when we consider both sanatoriums combined (*P* = 0.013 for a case fatality of 20.8%). No significant differences are found for Landeskogen sanatorium.

## 4. Discussion

While data on the 1918–1919 pandemic at these institutions are mostly limited to counts of cases and deaths, records indicate that infected employees or new patients brought the influenza virus to the sanatoriums, and then others were infected [29]. We hypothesized that patients and staff at the institutions would show no differences in morbidity or that morbidity would be highest for the patients, and that patients would have experienced higher case fatality. Results show that, contrary to our expectations, morbidity is generally lower among patients than the staff, with females, patients aged 10–39 years, and patients at Landeskogen having independently and significantly lower morbidity than the corresponding groups among employees. Our case fatality hypothesis is confirmed, although TB is significantly associated with influenza case fatality only for the age group 20–29 years for both sexes combined at Lyster sanatorium and for both sanatoriums combined, as well as for females aged 20–29 years when we consider both sanatoriums combined. The non-significant role of TB in male case fatality is likely a result of small sample size, as there are few male employee controls and few corresponding male ILI cases. Although we use a larger data set for both TB patients and employees than prior studies, the small male sample size is thus one weakness of our analysis. Results should be treated with caution, and results are not necessarily generalizable. Future work should therefore aim to study several TB sanatoriums, and to collect data across countries for cross-national comparisons.

Although TB patients may have had impaired immune systems and therefore were more susceptible to developing influenza once exposed [6,7,8,9,10,11,12,13], influenza morbidity in these institutional settings is generally lower for patients than for the employees. This result holds even after controlling for age, sex, and sanatorium. Our results thus add to the literature on the mixed findings on ILI in TB sanatoriums (e.g., [30]).

Observed morbidity differences in our sample might be due to differential contact patterns. Although we assumed that employees and patients would have no differences in crowding or mixing, the disease also apparently tended to cluster within certain rooms or wards [29]. Some of the employees also lived in their own residential housing, such as the doctors, nurses, stokers, and gardeners at Lyster. These employees thus may have lived under different crowding and assortative mixing conditions. Landeskogen, which used only 60% of the bed capacity in 1918, had significantly lower morbidity than Lyster, which had slightly more patients than they had beds. Separate models showed that patients had lower morbidity than employees only at Landeskogen, so the reduced amount of crowding at Landeskogen could therefore be the reason for lower morbidity overall. It is also likely that individuals infected with influenza were isolated, as described for the farm boy at Lyster and the employees at Landeskogen, while staff members would have come into contact with each other as well as with patients during their daily activities. Isolating patients may have been easier at Landeskogen, explaining why morbidity among patients was only lower there. 

Higher rates of disease among employees might also be explained by more frequent travel to the communities outside of the sanatoriums. However, both regions were rather remote and were not affected by the flu before the second national wave in the fall. At that time, there was low to very low mortality among the surrounding non-institutionalized population, making this explanation less likely. While PI mortality for the country as a whole in 1918 was 4.7 per 1000, in Bygland it was 4.2 per 1000 (SMR = 91.6, ns.) and in Lyster it was 1.6 per 1000 (SMR = 39.1, *p* < 0.05) [36]. Another confounding factor might be gender-related tasks of the employees (e.g., female nurses vs. male groundskeepers or maintenance workers) that would have influenced rates of contact, as well as potentially pre-existing health status. While we do not have information on the occupations of patients and employees at an individual level, the controls had a wide range of both low- and high-status occupations. Higher risk may be expected for employees directly engaged in patient interaction rather than other forms of employment. 

Underreporting of ILI cases in institutional settings, where patients would have easy access to doctor examinations, may not be as much of an issue compared to a study of the general population. We therefore believe that it is unlikely that the lower morbidity among patients is due to underreporting in this group. Because our analysis of fatality differences was conditioned on having an ILI, results are also not confounded by our finding that patients tended to have lower morbidity. We further assume that the results are only due to the negative impact of having a concomitant bacterial infection and not that the TB patients received poorer quality of basic health care than the employees did.

In general, the groups with a high risk of severe disease during the 1918 influenza pandemic were young adults (20–40 years), pregnant females, the previously ill, socially disadvantaged, and the indigenous Sami people [37,38,39,40]. In our analyses, we control for the observed differences in age, sex, and sanatorium in the multivariate ILI models and the same covariates are controlled for in the trivariate cross tables for case fatality. The age and sex profiles of the TB patients at the two sanatoriums were similar to the general Norwegian population of TB patients of the time; the typical patient was between 10 and 39 years and the highest mortality was for males in their 20s [41]. Additionally, the age-related pattern of high morbidity among young children and young adults and highest mortality among young adults observed in this sample matches those seen in the general population of Norway and elsewhere for the pandemic [38]. However, whether a patient had other diseases than TB, such as a heart or lung disease, kidney or liver disease, neurological disorders, or a compromised immune system, or whether a woman was pregnant, is unknown. The same goes for the employees, and we do not know whether employees had undetected TB infections. Studies have shown that health care workers, especially those working at in-patient facilities, are at higher risk for TB than the surrounding population [42,43]. As Lyster and Landeskogen were owned by the state, and stays were paid by the public, the patients probably also had a range of socioeconomic and occupational backgrounds. We do not have information on the ethnic status. However, we have no reason to believe that patients or employees were of Sami origin as most of the Sami people lived in the northernmost county of Finnmark, while Lyster was located in Sogn and Fjordane on the west coast and Landeskogen in the county of East-Agder in southern Norway. Considering these factors and that the two sanatoriums were among the larger ones in terms of bed capacity, it is likely that: (1) the sample patients are representative of the TB patients at other sanatoriums and facilities in the country, and (2) the sample employees are fairly representative of healthy controls without TB in the general Norwegian population. However, the institutionalized patients were most likely sicker than non-institutionalized TB sufferers were. We therefore cannot assume that results from institutions are representative for all persons with TB. 

Our consideration of co-infection with multiple pathogens as well as social factors including institutionalization reinforces the characterization of the 1918 pandemic as a complex syndemic. Our statistically significant findings for case fatality are consistent with previous research for sanatorium populations documenting a marginally significant impact of TB on case fatality during the 1918 pandemic as well as later influenza pandemics, particularly for prime-aged adults (e.g., [31]). Higher case fatality among TB patients than the employee controls is supportive of an active mechanism for the selection hypothesis as proposed by Noymer [24], although other potential contributors to mortality differences cannot be ruled out. Therefore, additional work on this topic is needed. Future work also should focus on continuing to tease apart the role of pre-existing health conditions and associated social factors, including potential stigma, in 1918 morbidity and mortality through analyses of data for populations that are institutionalized for a variety of health- and non-health-related reasons such as residents of boarding schools, prisons, asylums, and homes for the elderly or people with disabilities. 

## 5. Conclusions

The analyses presented here add to the literature base of the 1918 influenza pandemic by comparing morbidity and case fatality data at two sanatoriums in Norway, considering patients and staff controls as well as age and sex subgroups. Such sanatorium data are rare in general, but these records are particularly novel because they offer data on morbidity as well as mortality. Increased understanding of the role of TB as a risk factor in past pandemics is vital for contemporary public health preparedness planning. Tuberculosis remains a problem in many regions worldwide, especially developing countries, and as Mamelund [44] argues, social inequalities and vulnerable populations are under- or unaddressed in most pandemic preparedness plans. Through increased knowledge of the nuances of how TB and other pre-existing conditions might impact morbidity and mortality during future pandemics, such shortcomings in public health policy can be sufficiently addressed. 

## Figures and Tables

**Table 1 tropicalmed-04-00074-t001:** Distribution of covariates by case-control status (N and %).

Covariates	Cases: Tuberculosis (TB) Patients	Controls: Employees
N = sample population	201	97
Age (years)		
10–19	49 (24.38)	13 (13.40)
20–29	105 (52.24)	46 (47.42)
30–39	41 (20.40)	20 (20.62)
40–49	3 (1.49)	9 (9.28)
50–59	3 (1.49)	9 (9.28)
Average age	25.35	30.36
Sex		
Males	108 (53.73)	23 (23.71)
Females	93 (46.27)	74 (76.29)
Sanatorium		
Lyster	128 (63.68)	58 (59.79)
Landeskogen	73 (36.32)	39 (40.21)

**Table 2 tropicalmed-04-00074-t002:** Distribution of influenza-like illness (ILI) cases (% of pop) and ILI deaths (% of cases) by case-control status.

	TB Patients		Employees	
	Morbidity	Case Fatality	Morbidity	Case Fatality
All	84 (41.79)	15 (17.86)	62 (63.92)	0
Age (years)				
10–19	19 (38.78)	3 (15.79)	10 (76.92)	0
20–29	56 (53.33)	11 (19.64)	37 (80.43)	0
30–39	6 (14.63)	0	13 (65.00)	0
40–49	2 (66.67)	0	2 (22.22)	0
50–59	1 (33.33)	1 (100.00)	0 (00.00)	0
Sex				
Males	50 (46.30)	8 (16.00)	9 (39.13)	0
Females	34 (36.56)	7 (20.59)	53 (71.62)	0
Sanatorium				
Lyster	68 (53.13)	13 (19.12)	32 (55.17)	0
Landeskogen	16 (21.92)	2 (12.50)	39 (76.92)	0

**Table 3 tropicalmed-04-00074-t003:** Share and relative odds (with 95 percent confidence intervals) of having an ILI by TB status, age, sex, and sanatorium.

			Model 1	Model 2	Model 3	Model 4
Covariates	N (%) Not Sick	N (%) Sick	Unadj. OR (95% CI)	Adj. OR (95% CI)	Adj. OR (95% CI)	Adj. OR (95% CI)
Cases and controls						
TB-Patients	117 (58.2)	84 (41.8)	0.41 (0.25–0.67) *	0.25 (0.14–0.45) *	0.23 (0.12–0.43) *	0.22 (0.12–0.42) *
Employees	35 (36.1)	62 (63.9)	1.00 (ref.)	1.00 (ref.)	1.00 (ref.)	1.00 (ref.)
Age (years)						
10–19	33 (53.2)	29 (46.8)		1.00 (ref.)	1.00 (ref.)	1.00 (ref.)
20–29	58 (38.4)	93 (61.6)		1.68 (0.90–3.13)	1.70 (0.91–3.16)	1.97 (1.04–3.75) *
30–39	42 (68.8)	19 (31.2)		0.40 (0.18–0.88) *	0.40 (0.18–0.87) *	0.44 (0.20–0.98) *
40–49	8 (66.7)	4 (33.3)		0.25 (0.06–0.99) *	0.24 (0.06–0.96) *	0.26 (0.06–1.05)
50–59	11 (91.7)	1 (8.3)		0.04 (0.01–0.37) *	0.04 (0.00–0.35) *	0.04 (0.01–0.37) *
Sex						
Males	72 (55.0)	59 (45.0)			1.20 (0.71–2.04)	1.21 (0.70–2.07)
Females	80 (47.9)	87 (52.1)			1.00 (ref.)	1.00 (ref.)
Sanatorium						
Lyster	86 (46.2)	100 (53.8)				2.06 (1.20–3.54) *
Landeskogen	66 (58.9)	46 (41.1)				1.00 (ref.)
AIC			404.06	373.01	374.53	369.54
SC			411.45	395.19	400.41	399.12
−2 Log L			400.06	361.01	360.53	353.54
N=			298	298	298	298

* Sign. at 5% level.

**Table tropicalmed-04-00074-t004a:** a

Cases and Controls	ILI Cases	ILI Deaths	ILI Survivors	Case Fatality (ILI Deaths in % ILI Cases)	Statistical Test
Both sexes					
TB patients	68	13	55	19.12%	Fisher’s exact:
Employees	32	0	32	0.0%	*P* = 0.008
Males					
TB patients	37	6	31	16.22%	Fisher’s exact:
Employees	5	0	5	0.0%	*P* = 1.000
Females					
TB patients	31	7	24	22.58%	Fisher’s exact:
Employees	27	0	27	0.0%	*P* = 0.012

**Table tropicalmed-04-00074-t004b:** b

Cases and Controls	ILI Cases	ILI Deaths	ILI Survivors	Case Fatality (ILI Deaths in % ILI Cases)	Statistical Test
Both sexes					
TB patients	16	2	14	12.50%	Fisher’s exact:
Employees	30	0	30	0.00%	*P* = 0.116
Males					
TB patients	13	2	11	15.38%	Fisher’s exact:
Employees	4	0	4	0.00%	*P* = 1.000
Females					
TB patients	3	0	3	0.00%	-
Employees	26	0	26	0.00%	

**Table tropicalmed-04-00074-t004c:** c

Cases and Controls	ILI Cases	ILI Deaths	ILI Survivors	Case Fatality (ILI Deaths in % ILI Cases)	Statistical Test
Both sexes					
TB patients	84	15	69	17.86%	Chi square:
Employees	62	0	62	0.00%	*P* = 0.000
Males					
TB patients	50	8	42	16.00%	Fisher’s exact:
Employees	9	0	9	0.00%	*P* = 0.337
Females					
TB patients	34	7	27	20.60%	Fisher’s exact:
Employees	53	0	53	0.00%	*P* = 0.001

**Table tropicalmed-04-00074-t005a:** a

Age and Sex	ILI Cases	ILI Deaths	ILI Survivors	Case Fatality (ILI Deaths in % ILI Cases)	Statistical Test
Both sexes					
TB patients 10–19	17	2	15	11.76%	Fisher’s exact:
Employees 10–19	8	0	8	0.00%	*P* = 1.000
TB patients 20–29	42	10	32	23.81%	Fisher’s exact:
Employees 20–29	13	0	13	0.00%	*P* = 0.095
Females					
TB patients 10–19	7	2	5	28.57%	Fisher’s exact:
Employees 10–19	7	0	7	0.00%	*P* = 0.462
TB patients 20–29	22	5	17	22.72%	Fisher’s exact:
Employees 20–29	10	0	10	0.00%	*P* = 0.155
Males					
TB patients 10–19	10	0	10	0.00%	-
Employees 10–19	1	0	1	0.00%	
TB patients 20–29	20	5	15	25.0%	Fisher’s exact:
Employees 20–29	3	0	3	0.00%	*P* = 1.000
TB patients 50–59	1	1	0	100.00%	-
Employees 50–59	0	0	0	0.00%	

**Table tropicalmed-04-00074-t005b:** b

Age and Sex	ILI Cases	ILI Deaths	ILI Survivors	Case Fatality (ILI Deaths in % ILI Cases)	Statistical Test
Both sexes					
TB patients 10–19	2	1	1	50.00%	Fisher’s exact:
Employees 10–19	2	0	2	0.00%	*P* = 1.000
TB patients 20–29	14	1	13	7.14%	Fisher’s exact:
Employees 20–29	24	0	24	0.00%	*P* = 0.368
Females					
TB patients 10–19	1	0	1	0.00%	-
Employees 10–19	2	0	2	0.00%	
TB patients 20–29	2	0	2	0.00%	-
Employees 20–29	20	0	20	0.00%	
Males					
TB patients 10–19	1	1	0	100.00%	-
Employees 10–19	-	-	-	-	
TB patients 20–29	12	1	11	8.33%	Fisher’s exact:
Employees 20–29	4	0	4	0.00%	*P* = 1.000

**Table tropicalmed-04-00074-t005c:** c

Age and Sex	ILI Cases	ILI Deaths	ILI Survivors	Case Fatality (ILI Deaths in % ILI Cases)	Statistical Test
Both sexes					
TB patients 10–19	19	3	16	15.79%	Fisher’s exact:
Employees 10–19	10	0	10	0.00%	*P* = 0.532
TB patients 20–29	56	11	45	19.64%	Fisher’s exact:
Employees 20–29	37	0	37	0.00%	*P* = 0.003
Females					
TB patients 10–19	8	2	6	25.00%	Fisher’s exact:
Employees 10–19	9	0	9	0.00%	*P* = 0.206
TB patients 20–29	24	5	19	20.83%	Fisher’s exact:
Employees 20–29	30	0	30	0.00%	*P* = 0.013
Males					
TB patients 10–19	11	1	10	9.09%	Fisher’s exact:
Employees 10–19	1	0	1	0.00%	*P* = 1.000
TB patients 20–29	32	6	26	18.75%	Fisher’s exact:
Employees 20–29	7	0	7	0.00%	*P* = 0.568

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
