# Peer review of "Tuberculosis as a Risk Factor for 1918 Influenza Pandemic Outcomes"

_tropicalmed, 2019, doi:10.3390/tropicalmed4020074_

Reviewer 1 Report

This manuscript is well-written overall and presents a significant and valuable contribution to the literature on the relationship between tuberculosis and influenza morbidity and mortality, and on the factors responsible for the mortality patterns observed during the 1918 flu pandemic. This literature is often cited in work aiming to prepare for future pandemic influenza, and so has important practical value for public health today, as well as for broader historical understanding. However, I have a number of recommendations for revisions prior to publication.

I have had the opportunity to view the comments and recommendations of two other reviewers of the manuscript before finishing my own review, and I have to say that I agree with Reviewer 2 on their recommendation that the authors reconsider their statistical analysis approach in favour of logistic regression and odds ratios. As far as I have seen, those are the typical methods used in case-control analyses and so if the authors are to depart from that, I would expect to see some good justification for why another approach is more appropriate. As it stands, logistic regression with odds ratios would improve the quality and impact of the study – which despite weaknesses acknowledged by the authors in terms of sample size for some sub-groups, nevertheless represents a valuable contribution to the debate over the role of TB in influenza mortality during the 1918 pandemic, and its possible role today and in future pandemics. Regardless of exact methods used, I would like to see explicit statement of the decision-making process regarding statistical methods selection in the Materials and Methods section (i.e., description of why certain methods of analysis were selected for use with this data).

More minor revisions/corrections are needed in a few areas. Firstly, there is some confusion in how sex and gender are used throughout the manuscript, primarily in the tables. E.g., sometimes “male”/”female” is used, sometimes “men”/”women”, and sometimes “male gender”/”female gender”. The authors should use either sex or gender, and stick with consistent terminology accordingly (sex = male/female, gender = men/women).

Secondly, leading up to the statement of their main hypothesis (that there would be no difference in morbidity, but higher lethality for TB patients vs health controls), the authors describe three other “hypotheses”; I suggest that these would be better and more clearly framed (referred to) as “assumptions”, rather than as hypotheses.

Thirdly, I’m a bit confused by the statement on pg 15, lines 299-300 that “Not enough information is available to support whether these deaths were the result of active or passive selection mechanisms.” It was my understanding that this was part of the point of the case-control study: that by looking at lethality between the TB patients and the employees in the same age groups, finding a higher lethality among the TB patients would be evidence of active selection?

Other specific minor revisions needed:

-      Pg 1, line 38: the statement “Another influenza pandemic is imminent” needs some citations in support.

-      Pg 15, line 272: shouldn’t “somewhat higher morbidity for TB patients…” be “somewhat lower morbidity for TB patients…”?

Author Response

We are very happy to have the opportunity to revise and resubmit our paper. Below you will find the reviewer’s reports along with our answers in bold.

REVIEWER 1  

This manuscript is well-written overall and presents a significant and valuable contribution to the literature on the relationship between tuberculosis and influenza morbidity and mortality, and on the factors responsible for the mortality patterns observed during the 1918 flu pandemic. This literature is often cited in work aiming to prepare for future pandemic influenza, and so has important practical value for public health today, as well as for broader historical understanding. However, I have a number of recommendations for revisions prior to publication.

I have had the opportunity to view the comments and recommendations of two other reviewers of the manuscript before finishing my own review, and I have to say that I agree with Reviewer 2 on their recommendation that the authors reconsider their statistical analysis approach in favour of logistic regression and odds ratios. As far as I have seen, those are the typical methods used in case-control analyses and so if the authors are to depart from that, I would expect to see some good justification for why another approach is more appropriate. As it stands, logistic regression with odds ratios would improve the quality and impact of the study – which despite weaknesses acknowledged by the authors in terms of sample size for some sub-groups, nevertheless represents a valuable contribution to the debate over the role of TB in influenza mortality during the 1918 pandemic, and its possible role today and in future pandemics. Regardless of exact methods used, I would like to see explicit statement of the decision-making process regarding statistical methods selection in the Materials and Methods section (i.e., description of why certain methods of analysis were selected for use with this data).

-We have now done logistic regression models for ILI, but kept the cross-tables for the analysis of lethality as there were no deaths among the employees.

More minor revisions/corrections are needed in a few areas. Firstly, there is some confusion in how sex and gender are used throughout the manuscript, primarily in the tables. E.g., sometimes “male”/”female” is used, sometimes “men”/”women”, and sometimes “male gender”/”female gender”. The authors should use either sex or gender, and stick with consistent terminology accordingly (sex = male/female, gender = men/women).

-We now use sex=male/female throughout the paper.

Secondly, leading up to the statement of their main hypothesis (that there would be no difference in morbidity, but higher lethality for TB patients vs health controls), the authors describe three other “hypotheses”; I suggest that these would be better and more clearly framed (referred to) as “assumptions”, rather than as hypotheses.

-This is now changed, see bottom of page 2.

Thirdly, I’m a bit confused by the statement on pg 15, lines 299-300 that “Not enough information is available to support whether these deaths were the result of active or passive selection mechanisms.” It was my understanding that this was part of the point of the case-control study: that by looking at lethality between the TB patients and the employees in the same age groups, finding a higher lethality among the TB patients would be evidence of active selection?

-Thanks for this comment. We have now rephrased to “Higher lethality among TB patients than the employee controls is supportive of an active mechanism for the selection hypothesis as proposed by Noymer [14], although other potential contributors to mortality differences cannot be ruled out. Therefore, additional work on this topic is needed” (see lines 376-379)

Other specific minor revisions needed:

-      Pg 1, line 38: the statement “Another influenza pandemic is imminent” needs some citations in support.

-Thanks, new citations are include in line 39, refs 10 and 11.

-      Pg 15, line 272: shouldn’t “somewhat higher morbidity for TB patients…” be “somewhat lower morbidity for TB patients…”?

-This sentence is no longer in the paper.

Reviewer 2 Report

interesting addition to well-reviewed literature on excess mortality among TB infected during the pandemic 1918 'Spanish'  flu.

The number of tables is pretty long - don't know whether the journal uses electronic appendices.

The Conclusion is followed by extra or additional new text that in itself is not the conclusion . . please check this . 

lines 311/318 seem inappropriate as a conclusion.

minor points

line 117 . . . of which . . change into  . . of whom . .

line 243 . . association, NOT associaton . .

Author Response

We are very happy to have the opportunity to revise and resubmit our paper. Below you will find the reviewer’s reports along with our answers in bold.

REVIWER 2

interesting addition to well-reviewed literature on excess mortality among TB infected during the pandemic 1918 'Spanish'  flu.

-Thanks

The number of tables is pretty long - don't know whether the journal uses electronic appendices.

-We have now done a multivariate analysis for ILI substantially reducing the number of tables.

The Conclusion is followed by extra or additional new text that in itself is not the conclusion . . please check this . 

lines 311/318 seem inappropriate as a conclusion.

- The critique is somewhat vague as to what the objection is. However, we took out some text in the conclusion and tried to make the connections/arguments clearer. We don’t really agree that there’s new text, as it mostly returns to the pandemic preparedness mentioned in the intro, unless the reviewers objection is to the new reference.

minor points

line 117 . . . of which . . change into  . . of whom . .

-Thanks. We have now changed this.

line 243 . . association, NOT associaton . .

-Thanks. We have now changed this.

Reviewer 3 Report

This manuscript presents a study on morbidity and mortality due to influenza-like syndromes in TB-infected patients and controls. The authors use historical data extracted from records from two sanatoriums in 1918.

The dataset is appropriate, however data analysis is not. Instead of modelling morbidity and mortality depending on group, age, gender and sanatorium, the authors only compare their groups (employee versus patients) using chi-square and Fisher’s exact tests on a series of subsets extracted from the dataset. They thus test the same hypothesis on the whole dataset and on many subsets (data split by gender, age, sanatorium…). As a result, as many as 56 (!) chi-square and Fisher’s exact tests are performed, basically to test a single hypothesis. These tests are not independent, many of them are non-significant due to small sample size, and the overall risk of error is >>1. Moreover, the authors do not test anything else than the difference between employees and patients. In particular, they do not investigate whether confounding or interactions may occur. For example, the difference between patients and employees may depend on the sanatorium considered, age or gender. They do not consider the relationships between explanatory variables (e.g., age may differ between patients and employees, which may explain part of the differences observed). This analysis is not acceptable. For this reason, I considered that the quality of presentation of methods and results is not a relevant question, as the methods and results themselves are not relevant.

The basic way to analyse such data is logistic regression, the authors should seek help from a statistician or epidemiologist. Data should be summarized in a single table with variables “sanatorium”, "gender", "age", "group" (employees versus patients), "atrisk", "cases", "dead". Then a logistic model (I did it on grouped data) would show, for example, that there is a strong difference between employees and patients as for ILI morbidity, but also that this difference is not homogeneous according to sanatorium, age and gender. This analysis would 1) avoid multiple testing of the same hypothesis; 2) reinforce statistica test power; 3) give robust estimates of the effect sizes (given as odds-ratios, not probability difference which is meaningless), independent of other effects; 4) allow to test for other effects and interactions, thus bringing more detailed information.

The discussion does not consider alternative explanations for the differences between groups. In particular, patients are younger than employees (this can be tested very easily also), which may explain a part of the results obtained. The authors should discuss whether the sanatorium patients are representative of TB patients in general, and whether their two groups are comparable except for TB status. Is it possible that some employees may have undetected TB?

Minor points:

-          Line 38 “another influenza pandemic is imminent”: the authors should argue for such an assertion

-          The authors split a single dataset into multiple subset but not all subsets are formed. For example in table 7a on mortality, data on mortality in Lys sanatorium are given only for females aged 10-19 and 20-29, not for older age classes, whereas ILI cases were described in this class, with no explanation for that.

-          In some tables, figures are absent (for ex last line of Table 5b), does this mean “0”?

Author Response

We are very happy to have the opportunity to revise and resubmit our paper. Below you will find the reviewer’s reports along with our answers in bold.

REVIEWER 3

This manuscript presents a study on morbidity and mortality due to influenza-like syndromes in TB-infected patients and controls. The authors use historical data extracted from records from two sanatoriums in 1918.

The dataset is appropriate, however data analysis is not. Instead of modelling morbidity and mortality depending on group, age, gender and sanatorium, the authors only compare their groups (employee versus patients) using chi-square and Fisher’s exact tests on a series of subsets extracted from the dataset. They thus test the same hypothesis on the whole dataset and on many subsets (data split by gender, age, sanatorium…). As a result, as many as 56 (!) chi-square and Fisher’s exact tests are performed, basically to test a single hypothesis. These tests are not independent, many of them are non-significant due to small sample size, and the overall risk of error is >>1. Moreover, the authors do not test anything else than the difference between employees and patients. In particular, they do not investigate whether confounding or interactions may occur. For example, the difference between patients and employees may depend on the sanatorium considered, age or gender. They do not consider the relationships between explanatory variables (e.g., age may differ between patients and employees, which may explain part of the differences observed). This analysis is not acceptable. For this reason, I considered that the quality of presentation of methods and results is not a relevant question, as the methods and results themselves are not relevant.

The basic way to analyse such data is logistic regression, the authors should seek help from a statistician or epidemiologist. Data should be summarized in a single table with variables “sanatorium”, "gender", "age", "group" (employees versus patients), "atrisk", "cases", "dead". Then a logistic model (I did it on grouped data) would show, for example, that there is a strong difference between employees and patients as for ILI morbidity, but also that this difference is not homogeneous according to sanatorium, age and gender. This analysis would 1) avoid multiple testing of the same hypothesis; 2) reinforce statistica test power; 3) give robust estimates of the effect sizes (given as odds-ratios, not probability difference which is meaningless), independent of other effects; 4) allow to test for other effects and interactions, thus bringing more detailed information.

-Thanks so much for this suggestion. We have now done logistic regression models for ILI, but kept the cross-tables for the analysis of lethality as there were no deaths among the employees.

The discussion does not consider alternative explanations for the differences between groups. In particular, patients are younger than employees (this can be tested very easily also), which may explain a part of the results obtained. The authors should discuss whether the sanatorium patients are representative of TB patients in general, and whether their two groups are comparable except for TB status. Is it possible that some employees may have undetected TB?

 - Controlling for age substantially reduces the odds ratio for TB patients (from 0.41 to 0.25) because patients were younger and so more represented in age groups with higher morbidity. Subsequent controls for sex and sanatorium marginally reduce the odds ratio for TB patients, and in final model with all other factors the same, the morbidity is 88% lower than for employees.

-We have included a discussion on whether the pattern of morbidity and mortality in the sample resembles the same patterns in the general population; whether age and gender profile of the TB-patients (see also descriptive analysis) are representative of TB patients in other sanatoriums and TB patients outside sanatoriums; whether the two sanatoriums are representative in terms of size and bed-capacity (see materials and methods), and also a discussion on other potentitially unmeasured factors that may explain our findings (such as SES, pregnancy status, other illnesses than TB, ethnic status etc.) 

Minor points:

-          Line 38 “another influenza pandemic is imminent”: the authors should argue for such an assertion

-We replaced the old text with this: “Influenza pandemics occurs 3-4 times each century. After 1918, there were pandemics in 1957, 1968 and 2009. Most experts predict another influenza pandemic is not a matter of if, but when [10,11], so research on historical experiences, including the 1918 pandemic, provides vital insights to contemporary public health concerns” .

-          The authors split a single dataset into multiple subset but not all subsets are formed. For example in table 7a on mortality, data on mortality in Lys sanatorium are given only for females aged 10-19 and 20-29, not for older age classes, whereas ILI cases were described in this class, with no explanation for that.

-The new table 2 now clearly describes what age and genders that had ILI and deaths. The text describing the results for lethality is also clear on what groups that have deaths.  

-          In some tables, figures are absent (for ex last line of Table 5b), does this mean “0”?

-As we see it now, there are no more tables where figures are absent.

Reviewer 4 Report

This is an interesting paper and appreciate using a case-control study to examine a risk of disease for a health outcome, and that this type of data (sanitorium) is not commonly found/available. I think you should emphasize the uniqueness of the dataset, not just because you have age and sex information, but also morbidity is not common. You need to be more specific as to how this really improves our understanding of the relationship of TB and pandemic 1918 influenza, it would seem that you are supporting increased mortality in those with TB, but is there a novel finding or interpretation? Elaborating on the lower risk in TB patients could offer some new insight. A good rewrite of the manuscript is needed, I have offered some grammatical improvements, but there are more. Sentences are sometimes awkwardly phrased and overly wordy.

I have three major concerns.

1.     The relationship between TB and influenza is quite complex and simplifying it to TB increases risk of morbidity of influenza or death may not be meaningful, please provide a more detailed summary of the literature on this topic and recognize the limitations of using TB as the risk factor and not influenza as well.

2.     I find the results of lower morbidity in TB patients relative to the employees to be baffling, I can understand why no relationship exist as you proposed, (because of shared environment), but lower doesn’t make sense, in fact I would think another hypothesis would be increased morbidity considering the complex relationship/interaction between the two diseases. Most studies don’t have the chance to examine the morbidity interaction, so this should be mentioned. You say that there is some support in the literature for the lower morbidity, especially reference 20, please elaborate on this in the introduction. Also, a discussion of the limitations of morbidity influenza data is needed, especially with regards to underreporting, you could argue that because this study is in an institution setting underreporting may not be as problematic compared to a study on the general public. You offer population specific reasons in your discussion for lower/increase morbidity, such as employees having more interaction with the outside world, if this finding noted in other studies of similar institutions, back this up with the literature.

3.     I am not sure how meaningful the mortality results are, with the small sample sizes and no deaths in controls; the results are unstable. A change in one number (for example deaths in controls) can change the significance of the results from insignificant to significant. The results should be viewed with great caution and should be made clear that it isn’t generalizable.

Author Response

We appreciate the very thorough review and have attempted to address many of the points raised. Notably, we added material on the interactions between TB and flu, and on the larger declining pattern of TB in the 19th and 20th centuries. Several potential interpretations and limitations were raised and although we have not incorporated all of them, they did allow us to carefully consider these issues in more detail. This adds important context to the paper, and we are grateful for the suggested literature sources as well.

The results on morbidity do seem counterintuitive. We have acknowledged that higher morbidity is a reasonable expectation as well as our original hypothesis of no association. We have also attempted to clarify our discussion of these results, including potentially contradictory sentences on mixing patterns, occupations, etc. We also mention the possibility of underreporting, which is likely less of an issue for this sample.

We added details on the data source and reorganized the description of sanatoriums within the materials & methods section, as suggested.

We have addressed statistical concerns including tables regarding mortality, as well as noted that caution should be taken when interpreting the results.

Finally, we revised minor grammatical points as suggested or with slight variations.

Round  2

Reviewer 1 Report

I am pleased to see the attention the authors paid to the review comments, and I think the changes, especially some of the additional contextual information, have improved the paper.

One remaining issue that stands out in the revised paper (and which should be amended prior to acceptance/publication) is the statement on page 12, lines 363-364: "Although slightly more TB patients were male, significantly higher lethality was seen among female patients, which does not match sex biases observed in other studies [6]." It's not clear exactly what is referred to here; I'm assuming it is the percentages shown in Table 4c (male patient lethality 16%, female patient lethality 20.6%). However, male patient vs female patient lethality was not directly tested (and if it were, I think the authors would find that there is a non-significant difference), and so that statement needs to be revised. 

There are a few minor things for correction:

page 1, line 38: "Influenza pandemics occurs..." should read "Influenza pandemics occur...".

page 7, line 246: "6 men and 7 females" should read "6 males and 7 females".

page 8, Table 4c: the first statistical test is "Chi square", in contrast to every other test being Fisher's exact; I'm assuming the authors meant Fisher's exact here as well?

Author Response

We removed the sentence of concern in the first comment to avoid confusion about sex differences. We also addressed the minor corrections. Note that the Chi-square test is correct because the expected cells were large enough (above 5), and we have added text to clarify this.

Reviewer 3 Report

I guess that the authors have not realized that using logistic regression does not only mean adjusting models using logistic regression, but also make a complete analysis. Here the authors have added logistic models to their (already complex) data analysis, thus rendering it even more unreadable.

A modelling process would include comparing all biologically relevant models (with/without the effects of age, gender, sanatorium, TB status ….  + possible interactions), selecting the model that best fits the data (and is not too complex, using criteria like AIC or equivalent) and presenting the selected model(s), not all possible models. The steps of the analysis have to be explicited: which models were adjusted, with their results in terms of likelihood, number of parameters, AIC, weights… How did the authors choose one or several relevant models, and a complete description of the chosen model has to be presented.

In my previous report I suggested the authors to seek help of a statistician or data analysis, which they have not done apparently, but is definitely is necessary to analyse the data properly. There are arguments to suppose that the authors do not master data analysis, for example they argue that thy did not use logistic regression for mortality because not case of death was observed among employees ( which  is not e reason for not using logistic regression). Finally, as no model is explicitly written I guess that the authors have not adjusted models including interactions (not only additive models with possible confusion factors).

Author Response

We have consulted two statisticians and have added tests of model fit; steps of the analysis has been made explicit; it is now made clear which model that were unadjusted, adjusted, and for what factors. This also applies for the separate models. Logistic regression models for case fatality cannot be done because no deaths were observed among employees, as the attached example script from STATA shows.

Reviewer 4 Report

The analysis has been improved with logistical regression, however I still have major concerns with the mortality analysis.

As you point out because of the zero deaths in the one group, you can’t perform the logistical regression. Considering the limitations in the data, you cannot adequately do an analysis of mortality, and this should not be a focus of your paper. The paper should focus on morbidity only, remove the mortality (case fatality) from the paper or in the Discussion section you can mention the case fatality analysis but with great caution. Furthermore, for some age categories from the Landeskogen sanatorium (such as in table 5a), there are no deaths for both the patients and employees, why even bother showing a 0% case fatality ratio? This should be mentioned as a limitation and thrown out of the study. There is a space in the table following “employees 20-29”. Unfortunately, the mortality data without any deaths in the employees (and in some cases the patients) is not meaningful and doesn’t contribute to the study, especially if you add 1 death to your tables for the chi-square or fisher-s exact test this can change your results. The data is very unstable. Stating that caution needs to be taken in the discussion is good but doesn’t suffice, because you can’t adequately perform the analyses as suggested by other reviewers.

There needs to be a discussion about the study samples, especially the cases. Are the TB patients representative of the tuberculous from the general population? I would assume not, as these would be individuals who have had TB for a long time and now have a severe case of TB, hence why they are in a sanitorium. Perhaps they have also been in the sanitorium for a long time and are already at risk of dying. Do you have information on the duration of stay in the sanitorium and/or length or illness, or severity/stage of tuberculosis? For case-control studies, the cases and controls should represent as closely as possible respective individuals from the population.

I agree with reviewer’s 2 that differences in the age and sex composition between the two groups could account for differences in morbidity or mortality. This needs to be discussed in greater detail in the discussion.

Introduction

Reference 30, seems to mention tuberculosis and influenza in the context of institutions briefly in two sentences. Elaborate on their findings and provide evidence from other studies.

How exactly are you: “contribut[ing] to the understanding of the mechanisms leading to worse mortality outcomes for TB patients?” Explain and what mechanisms? Biological interactions? You are simply examining if TB patients experience co-morbidity and can’t get at the biomedical level of this co-morbidity.

You mention that your sample size is larger than other studies. What are these sample sizes and do they have morbidity and/or mortality data?

Line 97-98: Rewrite and remove the word “this” from the sentence.

Rephrase lines 101-102: “A study of the differential experience of the 2009 pandemic discovered….”

Rephrase lines 104-105: “In this paper we….” Or “In light of the dearth of research in…we examine…”

Lines 111-125: It is better to state that you hypothesize rather than assume. Can you indent the numbering (as one would with bullets? This would stand out more, and make it easier to read)

Line 119: Equally bad= ineffective

Line 123-124: “TB was associated with a higher influenza case fatality risk..” should be “higher risk of influenza death..”

Material & Methods

Line 137: The information on sanitoriums in 1935 doesn’t add to the paper. Is line 138 on 1935 as well? If so, it doesn’t help contextualize the study groups unless you can assume some of these characteristics were consistent throughout time and preceded 1935, then you must state this.

160- 172 this should be placed in a table, it is way too much for the reader to take in and digest

You need to describe the case fatality ratio in the methods. How did you calculate this value? (number of ILI deaths/number of ILI cases)

Line 265: “due to the few..”

Line 275: move “however” to after “males.”

Results

For table 3, place the information for TB cases first as TB is the risk factor and would be placed above the controls in the 2x2 table for the odds ratio. Is it possible to show the values for no flu? I think this might be more helpful to the reader than the total number of individuals.

Line 247: “.. is 88% lower than for employees.” should be, “… is 88% lower than the employee rate.”

Discussion

The sentence on line 308 is awkward.

Line 347-348: What does the… “got the disease late” mean? Rewrite this statement

General writing comments

The paper needs more editing. There are other instances of grammatical problems. Please avoid the overuse of “do”, “be” and “got,” and be mindful of tenses, such as pastvs. present tense.

Author Response

The analysis has been improved with logistical regression, however I still have major concerns with the mortality analysis.

As you point out because of the zero deaths in the one group, you can’t perform the logistical regression. Considering the limitations in the data, you cannot adequately do an analysis of mortality, and this should not be a focus of your paper. The paper should focus on morbidity only, remove the mortality (case fatality) from the paper or in the Discussion section you can mention the case fatality analysis but with great caution.

-Thanks for your comment. We have chosen to keep the mortality analysis as we are of the opinion that it is a substantial contribution to the existing literature. We have added text to highlight that our sample is 2-3 times larger than previous work from Oei and Nishiura, making it possible to reinforce their finding of a marginally significant impact of TB as well as provide new analyses showing a significant effect at 5% or lower for our subgroups considering sex, age and sanatorium residence. These findings are also on par with prior research showing that those in their late 20s were most at risk during the 1918 influenza pandemic, further lending support to our results

Further, for these analyses, we use the Fisher’s exact test, which is designed especially for small samples. The lack of significant case-control differences in case fatality for males, likely due to few male deaths and few male employee controls, suggests this test works as would be expected with our small samples.

Furthermore, for some age categories from the Landeskogen sanatorium (such as in table 5a), there are no deaths for both the patients and employees, why even bother showing a 0% case fatality ratio? This should be mentioned as a limitation and thrown out of the study. There is a space in the table following “employees 20-29”. Unfortunately, the mortality data without any deaths in the employees (and in some cases the patients) is not meaningful and doesn’t contribute to the study, especially if you add 1 death to your tables for the chi-square or fisher-s exact test this can change your results. The data is very unstable. Stating that caution needs to be taken in the discussion is good but doesn’t suffice, because you can’t adequately perform the analyses as suggested by other reviewers.

- Our results show that these data need to be disaggregated by age, gender and sanatorium in order to separate significant effects from non-significant effects. As we now explain at the bottom of section 2.2., “In order to be fully transparent with our results, we present each cell even if they contain the value 0 (no ILI cases or deaths) or a “-“, meaning that there were no TB patients or employees in this category”.

There needs to be a discussion about the study samples, especially the cases. Are the TB patients representative of the tuberculous from the general population? I would assume not, as these would be individuals who have had TB for a long time and now have a severe case of TB, hence why they are in a sanitorium. Perhaps they have also been in the sanitorium for a long time and are already at risk of dying. Do you have information on the duration of stay in the sanitorium and/or length or illness, or severity/stage of tuberculosis? For case-control studies, the cases and controls should represent as closely as possible respective individuals from the population.

-Thanks for this comment. In 2.1, we now state that “We do not have information on the illness onset, illness severity, or the durations of patient stays at the sanatoriums. However, it is reasonable to believe that those institutionalized were sicker than the non-institutionalized suffering from tuberculosis”. In the discussion we state that “However, the institutionalized patients were most likely sicker than non-institutionalized TB sufferers were. We therefore cannot assume that results from institutions are representative for all persons with TB”.

I agree with reviewer’s 2 that differences in the age and sex composition between the two groups could account for differences in morbidity or mortality. This needs to be discussed in greater detail in the discussion.

-We are controlling for age and sex in our morbidity analysis, and we note in several places (in the results and discussion sections) that the results hold even after these controls were made. In other words, age and sex composition did not account for morbidity differences. The trivariate analysis for lethality also controls for age and sex.

Introduction

Reference 30, seems to mention tuberculosis and influenza in the context of institutions briefly in two sentences. Elaborate on their findings and provide evidence from other studies.

-Thanks again for your comment. We added some details incl. that the data are from the USA, that no significance tests were done by data gathered by Jordan in his review etc., see new additions in the introduction.

How exactly are you: “contribut[ing] to the understanding of the mechanisms leading to worse mortality outcomes for TB patients?” Explain and what mechanisms? Biological interactions? You are simply examining if TB patients experience co-morbidity and can’t get at the biomedical level of this co-morbidity.

-We replaced the old text with this: “Comparing the differences in case fatality rather than mortality (deaths/population) enables us to control for the observed differential risk of developing an ILI given exposure. We thus contribute to the understanding of the circumstances leading to worse outcomes for TB patients”.

You mention that your sample size is larger than other studies. What are these sample sizes and do they have morbidity and/or mortality data?

-We now write this in the introduction: “These analyses are made possible by a sample size that is 2-3 times larger than prior studies analyzing data from TB sanatoriums, such as Oei and Nishiura [31], who had data for only one sanatorium with 102 TB patients and 33 employees. In our study, we have data for two sanatoriums and report results by age and sex for 201 TB patients and 97 employees”.

-We also cite Oei and Nishiura before this section, showing that they had information on both morbidity and mortality.

Line 97-98: Rewrite and remove the word “this” from the sentence.

Rephrase lines 101-102: “A study of the differential experience of the 2009 pandemic discovered….”

Rephrase lines 104-105: “In this paper we….” Or “In light of the dearth of research in…we examine…”

Lines 111-125: It is better to state that you hypothesize rather than assume. Can you indent the numbering (as one would with bullets? This would stand out more, and make it easier to read)

- This is the exact opposite recommendation of an earlier review. We agree with that one (to use assume rather than hypothesize as we did in a prior draft) because these are not research hypotheses that we have data to test in the analyses, and so hypothesize would potentially be confusing. Because of that, we also believe that indenting them would draw more attention to them and make it seem more important than they are.

Line 119: Equally bad= ineffective

Line 123-124: “TB was associated with a higher influenza case fatality risk..” should be “higher risk of influenza death..”

Material & Methods

Line 137: The information on sanitoriums in 1935 doesn’t add to the paper. Is line 138 on 1935 as well? If so, it doesn’t help contextualize the study groups unless you can assume some of these characteristics were consistent throughout time and preceded 1935, then you must state this.

-Thanks. We have rephrased this section.

160- 172 this should be placed in a table, it is way too much for the reader to take in and digest

-We’ve already been told to minimize the number of tables in previous drafts. We have chosen to shorten this section.

You need to describe the case fatality ratio in the methods. How did you calculate this value? (number of ILI deaths/number of ILI cases)

-Thanks. This is now stated clearly several times, including in the methods section.

Line 265: “due to the few..”

-Thanks. Done.

Line 275: move “however” to after “males.”

-Thanks. Done.

Results

For table 3, place the information for TB cases first as TB is the risk factor and would be placed above the controls in the 2x2 table for the odds ratio. Is it possible to show the values for no flu? I think this might be more helpful to the reader than the total number of individuals.

 -Thanks. Done.

Line 247: “.. is 88% lower than for employees.” should be, “… is 88% lower than the employee rate.”

 -Thanks. Done.

Discussion

 The sentence on line 308 is awkward.

Line 347-348: What does the… “got the disease late” mean? Rewrite this statement

 -These sentences and statements are rewritten.

General writing comments

The paper needs more editing. There are other instances of grammatical problems. Please avoid the overuse of “do”, “be” and “got,” and be mindful of tenses, such as pastvs. present tense.

-Thanks. We have done our best in terms of editing.

Round  3

Reviewer 4 Report

The manuscript is improved and can now be accepted for publication.